# In-Silico Evaluation of Anthropomorphic Measurement Variations on Electrical Cardiometry in Neonates

**DOI:** 10.3390/children8100936

**Published:** 2021-10-18

**Authors:** David B. Healy, Eugene M. Dempsey, John M. O’Toole, Christoph E. Schwarz

**Affiliations:** 1Department of Neonatology, Cork University Maternity Hospital, Wilton, T12 K8AF Cork, Ireland; david.healy@ucc.ie (D.B.H.); g.dempsey@ucc.ie (E.M.D.); jotoole@ucc.ie (J.M.O.); 2Department of Paediatrics & Child Health, University College Cork, T12 K8AF Cork, Ireland; 3INFANT Research Centre, Wilton, T12 K8AF Cork, Ireland; 4Department of Neonatology, University Children’s Hospital, 72076 Tübingen, Germany

**Keywords:** infant, premature, term, bio-impedance, non-invasive cardiac output monitoring

## Abstract

Non-invasive cardiac output methods such as Electrical Cardiometry (EC) are relatively novel assessment tools for neonates and they enable continuous monitoring of stroke volume (SV). An in-silico comparison of differences in EC-derived SV in relation to preset length and weight was performed. EC (ICON, Osypka Medical) was simulated using the “demo” mode for various combinations of length and weight representative of term and preterm infants. One-centimetre length error resulted in a SV-change of 1.8–3.6% (preterm) or 1.6–2.0% (term) throughout the tested weight ranges. One-hundred gram error in weight measurement resulted in a SV-change of 5.0–7.1% (preterm) or 1.5–1.8% (term) throughout the tested length ranges. Algorithms to calculate EC-derived SV incorporate anthropomorphic measurements. Therefore, inaccuracy in physical measurement can impact absolute EC measurements. This should be considered in the interpretation of previous findings and the design of future clinical studies of EC-derived cardiac parameters in neonates, particularly in the preterm cohorts where a proportional change was noted to be greatest.

## 1. Introduction

Neonatal non-invasive evaluation of cardiac output (CO) is typically performed with echocardiography. Electrical cardiometry (EC) enables a continuous assessment of CO in newborns. Parameters, including stroke volume (SV) and, multiplied by heart rate (HR), cardiac output (CO), are recorded as absolute values or indexed for body weight. While EC is now increasingly used in clinical research in neonates [1], including the delivery room and neonatal intensive care unit [1,2,3,4,5], further research and development of the technology is required before it can be appropriately utilized in clinical practice to potentially guide management.

In EC, the distance between electrodes influences the recorded parameters and, as such, an initial calibration using weight and length is required. Therefore, inaccurate anthropomorphic measurements might affect EC-derived parameters. Measurement is particularly challenging in the delivery room, and in previous studies, predetermined standardized measurements were employed [2,3,4,5]. Inaccuracy may become significant where absolute values are used to delineate ‘normative ranges’. Thus, our aim was to evaluate the effect size of differences in length and weight on EC-derived SV estimates for preterm and term infants in a simulation study.

## 2. Materials and Methods

The ICON monitor (Osypka Medical, Berlin, Germany) was used in “demonstration mode”, which provides an identical set of electrical impedance changes for each repetition representative for an adult patient. As sex does not have an effect on the measurements calculated by the ICON monitor (unpublished data and confirmed by the manufacturer), all measurements were performed as for a male on day 1 postnatally. Weight can be set to increments of 25 g (<1 kg) or 50 g (>1 kg) and length can be input in increments of 0.1 cm on the monitor. We used a representative range of weights and lengths (term: weight 3.0–4.2 kg, length 48–56 cm/preterm: weight 0.5–1.5 kg, length 28–40 cm). SV for each combination of weight and length was documented after a 2-min cycling period. A linear regression model was fitted to the data to find an analytical expression for the dependency of SV on weight and length. Models were developed separately for the term and preterm data using an ordinary least squares fit (Python 3.9 with statsmodel 0.12). The dependent variable was SV. Potential independent variables were weight [kg], length [cm], and the weight-by-length interaction. Models with all combinations of the independent variables were compared using the Akaike information criterion.

## 3. Results

Appendix A illustrates the dependence of SV on weight and length. For a term infant with a length of 51 cm, every 100-g difference from the median weight of 3.5 kg led to a change in SV of 1.7%. For preterm infants, this effect was found to be more influential on the resultant SV measurement; e.g., for a 34-cm infant, each 100-g increase or decrease from 1 kg led to a relative change in measured SV of 6.1%. In contrast every 1-cm change in length resulted in a 1.8% relative change in SV for median weight term infants and 2.7% for median weight preterm infants.

The SV regression models for the preterm and term data both included length and weight-by-length as independent variables (see Table 1). Relative SVs, normalized to the median of the variable, are presented in Appendix A.

## 4. Discussion

We quantified the effect of weight and length on EC-derived SV estimates. Differences in anthropometric measures, and thereby, differences in intrathoracic intravascular volumes, lead to differences in SV measurements. The effects were more pronounced in preterm infants compared to term infants. Therefore, inaccurate estimates or measurements of these anthropometric values can lead to clinically relevant changes in SV.

While, in adults, such a small proportional error makes a clinically irrelevant difference, in infants, errors in anthropomorphic measurement could lead to significant discrepancies in SV. Furthermore, as CO equals SV times HR, the effect of the naturally higher newborn HR on CO estimation leads to amplification of the initial weight or length error. For example, from our simulation data, overestimation of a preterm infant’s weight by 250 g and their length by 2 cm would subsequently result in a 0.33-mL error in SV. Based on a HR of 160 bpm in a 1-kg neonate, this would result in an error of almost 53 mL/s/kg/min in CO. In term infants, however, the relative error in SV is lower, and the amplification of the error is less pronounced due to slightly lower HR. Therefore, for a 200-g weight and a 2-cm length overestimation, the calculated error in CO would be 12.4 mL/s/kg/min for a 3.4-kg neonate.

Even the error in a single measurement (i.e., weight or length) can still lead to relevant change. For example, a 2-cm measurement inaccuracy would lead to almost 15 mL/s/kg/min in CO error for a 1-kg baby. The effect of length as a component in the algorithm calculation is important and raises important clinical concerns, as accurate measurement in neonates is not always straightforward [6]. Due to the expected inaccuracy in length, it is common in neonatology to index output by weight and not by body surface area. However, as EC parameters depend on the distance between outer and inner electrodes, calibration with length is necessary to adjust for differences in distance caused by body length. The ability to input weight only to the closest 25 g should also be considered as a factor when evaluating cardiac indices in extremely preterm infants, where seemingly small weight differences may also lead to clinically relevant distinctions in CO. This is particularly the case for the smallest preterm infants, in whom 25 g can represent up to 5% of a weight difference.

Accounting for this, researchers should be aware of the potential for algorithmic reproduction of weight and length estimation errors resulting in “abnormal” EC-derived parameters. Noori et al. mentioned calibration within the EC algorithm in 2012 [7]. However, few studies since then have reported how calibration for weight or length was addressed. This is pertinent in the delivery room. Previous studies used pre-set or standardized weight and length for term infants in the delivery setting [2,3,4,5]. In one study, EC measurements were performed during intact cord management, therefore relying solely on estimates [2]. As this will result in miscalibration, EC-derived parameters should only be used to provide trends, until weight and length are verified. Of note, only flow related measurements are affected (i.e., SV or CO), whereas other EC parameters are not (e.g., thoracic fluid content). While we would not currently recommend the use of EC for the determination of absolute CO values to guide clinical management decisions yet, it may be possible to improve accuracy in the delivery room research setting. Use of resuscitaires with in-built weighing scales could rapidly provide more accurate estimates of weight, or, where these are not available, the equipment could be pre-set based on the average weight for the gestational age or by using the estimated fetal weight derived from antenatal ultrasound scans. Length could be quickly estimated by employing a fixed ruler attached to the resuscitaire, without significant interference to clinical management of the baby.

When used in the research setting, it is feasible that correct measurements could be used in conjunction with raw data during analysis to produce more accurate estimates. However, without in-depth knowledge of the algorithm, this could have unreliable results. To enable the accurate investigation of cardiac parameters in the early transitional period and improve technological development, manufacturers could integrate an option for post-hoc correction of data based on reliable anthropomorphic measurements.

## 5. Limitations

This study is an *in-silico* study and does not reflect the complexity and variability of delivery room situations or other real-life scenarios. However, this enables high comparability as the parameters obtained by EC always use the same data sets of electrical impedance changes and HR; therefore, only the pre-set parameters affected changes in SV values. As the simulation is based on adult data, the absolute values of CO should not be interpreted and only the changes incurred by the alteration of weight and length are meaningful. The algorithm itself, including the method of calibration is kept confidential by the manufacturer. Inclusion of the weight–length interaction term in the linear-regression modelling indicates a nonlinear relation between weight, length, and SV. Plots of raw data in Appendix A, however, do suggest that the dominant relation is linear. Therefore, this *in-silico* study provides information to users on the effect size and, by this, the relevance of accurate weight and length measurement for EC.

## Figures and Tables

**Table 1 children-08-00936-t001:** Coefficients of linear regression model of stroke volume.

	Preterm Model Coefficient (95% CI)	Term Model Coefficient (95% CI)
intercept [mL]	0.1503 (0.079 to 0.221)	0.4801 (0.423 to 0.537)
length [cm]	0.0141 (0.012 to 0.016)	0.0298 (0.029 to 0.031)
weight × length [kg × cm]	0.0285 (0.028 to 0.029)	0.0160 (0.016 to 0.016)

## Data Availability

All relevant data generated or analyzed during this study are included in this article or its supplementary material. Further enquiries can be directed to the corresponding author.

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
