# Peer review of "In-Silico Evaluation of Anthropomorphic Measurement Variations on Electrical Cardiometry in Neonates"

_children, 2021, doi:10.3390/children8100936_

Round 1

Reviewer 1 Report

This short communication written by Healy et al. is about the in-silico evaluation of measurement variations on electrical cardiometry in neonates. Monitoring cardiac output in neonates, especially continuously and non-invasively is an important topic in NICU. This article is only based on in-silico evaluation, but might be clinically relevant and interesting. I have just few comments.

  • Based on these evaluation data, what would the authors suggest to do for the measurements in the delivery room? To get the data during the first minutes of life, where the interesting changes happen in the cardiovascular system, the pre-setting must be already prepared before the birth. And it is not possible to get the exact anthropomorphic data of the neonates.

  • Is there any possibility to correct the data properly (for example with a standardized mathematical formula) according to the exact anthropomorphic data after the measurement?

  • As it is mentioned in the limitation the simulation was based on adult data and the final formula is unknown, is it possible or even reasonable to claim these “length and weight dependent” differences in SV and CO in general?

Author Response

Reviewer1

 Summary

This short communication written by Healy et al. is about the in-silico evaluation of measurement variations on electrical cardiometry in neonates. Monitoring cardiac output in neonates, especially continuously and non-invasively is an important topic in NICU. This article is only based on in-silico evaluation, but might be clinically relevant and interesting. I have just few comments.

  • Based on these evaluation data, what would the authors suggest to do for the measurements in the delivery room? To get the data during the first minutes of life, where the interesting changes happen in the cardiovascular system, the pre-setting must be already prepared before the birth. And it is not possible to get the exact anthropomorphic data of the neonates.

Response: Thank you for your comment. This is the primary issue with using this technology in the delivery room, however there may be ways to improve accuracy of weight measurement in the delivery room. Use of estimated fetal weights derived from the most recent fetal ultrasound could provide a reasonable estimate for initial setup. Alternatively, researchers could use average weight and length for gestational age ranges rather than using a “one measurement for all” approach. Where available utilization of resuscitaires with in-built weighing scales could provide rapid estimation of weight in order to then quickly set the monitor while electrodes are being applied to the infant. We have added a short paragraph reflecting these recommendations at the end of the discussion.

“While we would not currenlty recommend the use of EC for determination of absolute CO values to guide clinical management decisions, it may be possible to improve accuracy in the delivery room research setting. Use of resuscitaires with in-built weighing scales could rapidly provide more accurate estimates of weight, or, where these are not available, equipment could be pre-set based on average weight for gestational age or by using the estimated fetal weight derived from antenatal ultrasound scans. Length could be quickly estimated by employing a fixed ruler attached to the resuscitaire, without significant interference to clinical management of the baby.”

  • Is there any possibility to correct the data properly (for example with a standardized mathematical formula) according to the exact anthropomorphic data after the measurement?

Response: This is an important query. As the technology currently stands this would not be possible to do with definite accuracy without in depth knowledge of the algorithm. Going forward, this could be an option that the manufacturers of such devices might integrate into their monitors (i.e. retrospective correction of estimates). We have added a short paragraph to the end of the discussion to reflect this.

“When used in the research setting, it is feasible that correct measurements could be used in conjunction with raw data during analysis to produce more accurate estimates. However, without in-depth knowledge of the algorithm this could have unreliable results. To enable accurate investigation of cardiac parameters in the early transitional period and improve technological development manufacturers could integrate an option for post-hoc correction of data based on reliable anthropomorphic measurements.”

  • As it is mentioned in the limitation the simulation was based on adult data and the final formula is unknown, is it possible or even reasonable to claim these “length and weight dependent” differences in SV and CO in general?

Response: Thank you for this question. The ICON monitor provides a demonstration mode that uses an adult-derived cycle of electrical signal, which ultimately provides absolute output values that are not clinically relevant to neonates. However, regardless of the morphology of the input signal, alteration of weight and length produce proportional changes in the output. Data from the input signal is only one element in the algorithm for calculation of SV and CO but weight and length appear to also be important components in that algorithm. Our study is the first to describe this effect. The proportional change is the relevant result in this study which would still be the result were an “infant appropriate” cardiac cycle used here. We have clarified the sentence in the “Limitations” section.

“As the simulation is based on adult data, the absolute values of CO should not be interpreted and only the changes incurred by alteration of weight and length are meaningful.”

Reviewer 2 Report

This study uses the simulation model to investigate the impact of incorrect height and weight on the cardiac output measurement by non-invasive evaluation method. This study result will provide a preliminary simulation data to remind the importance of anthropomorphic measurement variation during clinical practice. I have some suggestions for the authors and hope that my comments are constructive to them.

  1. During resuscitation process in the delivery room, the use of this machine is not suitable for the immediate response in a short time. We can’t wait for 2 minutes to get the cardiac output information and then to next step for resuscitation. It is suggested that the author can strengthen the situation under which condition that the height and weight cannot be accurately measured, and the data of cardiac output is urgently needed. Or you can consider adding the discussion about the impact of the height and weight measurement errors of premature infants in the clinical practice and how it could impact the cardiac measurement outcomes.
  2. Page 2, Line 46-47. The author mentioned “as sex does not have an effect on the measurements…” However, in your reference #5, Baik-Schneditz et al. already reported that sex related differences in cardiac output in neonates during postnatal transition. They found that a significantly higher cardiac output in male neonates. Please discuss briefly if gender also affects the measurement, how this might change your research findings.

Author Response

Reviewer 2

This study uses the simulation model to investigate the impact of incorrect height and weight on the cardiac output measurement by non-invasive evaluation method. This study result will provide a preliminary simulation data to remind the importance of anthropomorphic measurement variation during clinical practice. I have some suggestions for the authors and hope that my comments are constructive to them.

  1. During resuscitation process in the delivery room, the use of this machine is not suitable for the immediate response in a short time. We can’t wait for 2 minutes to get the cardiac output information and then to next step for resuscitation. It is suggested that the author can strengthen the situation under which condition that the height and weight cannot be accurately measured, and the data of cardiac output is urgently needed. Or you can consider adding the discussion about the impact of the height and weight measurement errors of premature infants in the clinical practice and how it could impact the cardiac measurement outcomes.

Response: Thank you for this comment. Use in the delivery room is one of the most challenging hurdles for this technology in neonates. We have added some suggestions at the end of the discussion as to how weight estimation could be improved for delivery room studies that are utilising EC. However, it is important to note that this technology is not yet reliable enough for use in clinical practice of neonatology, even were the inaccuracies around anthropomorphic measurements resolved.

“While we would not currently recommend the use of EC for determination of absolute CO values to guide clinical management decisions yet, it may be possible to improve accuracy in the delivery room research setting. Use of resuscitaires with in-built weighing scales could rapidly provide more accurate estimates of weight, or, where these are not available, equipment could be pre-set based on average weight for gestational age or by using the estimated fetal weight derived from antenatal ultrasound scans.“

  1. Page 2, Line 46-47. The author mentioned “as sex does not have an effect on the measurements…” However, in your reference #5, Baik-Schneditz et al. already reported that sex related differences in cardiac output in neonates during postnatal transition. They found that a significantly higher cardiac output in male neonates. Please discuss briefly if gender also affects the measurement, how this might change your research findings.

Response: Thank you for your comment. You are correct that reference number 5 reports sex-differences in cardiac output in neonates. Our statement was referring to the fact that when using the ICON monitor choosing male or female during set up makes no difference to the output values, as this factor is not a component in their algorithm (as confirmed by Osypka). This sentence has been clarified in the text.

“As sex does not have an effect on the measurements calculated by the ICON monitor (unpublished data and confirmed by the manufacturer), all measurements were performed as for a male on day 1 postnatally.”